# Transcriptome analysis of female western flower thrips, *Frankliniella occidentalis*, exhibiting neo-panoistic ovarian development

**Du-Yeol Choi, Yonggyun Kim** *

Department of Plant Medicals, College of Life Sciences, Andong National University, Andong, Korea

* hosanna@anu.ac.kr

**Data Availability Statement:** All transcript files are available from the GenBank database (accession number: PRJNA833754).

## Abstract

The western flower thrips, *Frankliniella occidentalis*, is one of the most devastating insect pests with explosive reproductive potential. However, its reproductive physiological processes are not well understood. This study reports the ovarian development and associated transcriptomes of *F. occidentalis*. Each ovary consisted of four ovarioles, each of which contained a maximum of nine follicles in the vitellarium. The germarium consisted of several dividing cells forming a germ cell cluster, presumably consisting of oocytes and nurse cells. The nurse cells were restricted to the germarium while the subsequent follicles did not possess nurse cells or a nutritive cord, supporting the neo-panoistic ovariole usually found in thysanopteran insects. Oocyte development was completed 72 h after adult emergence (AAE). Transcriptome analysis was performed at mid (36 h AAE) and late (60 h AAE) ovarian developmental stages using RNA sequencing (RNASeq) technology. More than 120 million reads per replication were matched to ≈ 15,000 *F. occidentalis* genes. Almost 500 genes were differentially expressed at each of the mid and late ovarian developmental stages. Kyoto Encyclopedia of Genes and Genomes (KEGG) analysis showed that these differentially expressed genes (DEGs) were associated with metabolic pathways along with protein and nucleic acid biosynthesis. In both ovarian developmental stages, vitellogenin, mucin, and chorion genes were highly (> 8-fold) expressed. Endocrine signals associated with ovarian development were further investigated from the DEGs. Insulin and juvenile hormone signals were upregulated only at 36 h AAE, whereas the ecdysteroid signal was highly maintained at 60 h AAE. This study reports the transcriptome associated with the ovarian development of *F. occidentalis*, which possesses a neo-panoistic ovariole.

## 1. Introduction

The western flower thrip, *Frankliniella occidentalis* (Pergande) (Thysanoptera: Thripidae), is one of the most devastating insect pests to many horticultural crops, especially those in greenhouses [1]. Both the larval and adult stages cause damage to plants by directly feeding on leaves or flowers. Especially, adults transmit plant viruses including tomato spotted wilt virus (TSWV) [2]. TSWV infection becomes serious and causes massive economic loss in hot

**Funding:** This work was performed with the support of the "Cooperative Research Program for Agriculture Science & Technology Development (Project No. PJ01578901)" funded by the Rural Development Administration, Republic of Korea. The funder had no role in study design, data collection and analysis, decision to publish, or preparation of the manuscript.

**Competing interests:** The authors have declared that no competing interests exist.

pepper production in Korea [3]. This pest, originally native to North America, has spread to more than 60 countries since the late 1970s, including Canada, Australia, the United Kingdom, and far East Asian countries [4].

Various techniques such as chemical insecticides, entomopathogens, and pheromone traps have been applied to control *F. occidentalis* without satisfactory efficacy due to the insect's specific hiding behavior and insecticide resistance [5]. The thrips exhibits arrhenotokous parthenogenesis, with females developing from fertilized eggs and males from unfertilized eggs [6]. A brief immature period less than 10 days along with this various reproductive modes allow the thrips to rapidly build up the field populations during crop cultivating periods and so frequently leads to outbreaks beyond economic injury level [6]. Along with high reproductive potential, this type of mating behavior contributes to a rapid population increase and the development of insecticide resistance [7]. However, the molecular processes underlying reproduction and its regulation in this species remain unclear.

To investigate the physiological processes of the ovarian development of *F. occidentalis*, transcriptome analysis is useful for understanding the expressed genes associated with reproduction. A draft genome (415.8 Mb) of *F. occidentalis* was sequenced and its 16,859 genes were annotated into different functional categories including chemosensory receptors, detoxification, salivary gland, immunity, and development [8]. This suggests that RNA sequencing (RNASeq) analysis would be highly validated by this genomic information.

To identify the genes associated with ovarian development, this study investigated the ovarian development of *F. occidentalis* after adult emergence. After determining the mid and late ovarian developmental stages, transcriptomes were assessed using the NovaSeq 6000 platform. Subsequent differentially expressed gene (DEG) analysis in different developmental stages of female adults predicted the genes associated with ovarian development.

## 2. Materials and methods

### 2.1. Thrip rearing

*F. occidentalis* adults were obtained from Bio Utility, Inc. (Andong, Korea) and reared in a laboratory under conditions of 27 ± 1˚C constant temperature, a 16:8 h (light:dark) photoperiod, and relative humidity of 60 ± 5%. The insects were reared on sprouted bean seed kernels.

### 2.2. Dissection of ovaries and microscopic observation

Different ages of female western flower thrips were dissected in 1 × phosphate-buffered saline (PBS) under a stereomicroscope at 30× magnification. PBS was prepared with 100 mM phosphate buffer containing 0.7% NaCl (pH 7.4). The ovaries were pulled from the abdominal tip and fixed with 3.7% paraformaldehyde in a wet chamber under darkness at room temperature (RT) for 60 min. After washing three times with 1 × PBS, the cells in the ovarioles were permeabilized with 0.2% Triton X-100 in 1 × PBS at RT for 20 min. The cells were then washed three times and blocked with 5% skim milk (MB cell, Seoul, Korea) in 1 × PBS at RT for 60 min. After washing three times, the cells were incubated with DAPI (4′,6-diamidino-2-phenylindole, 1 mg/mL) diluted 1,000 times in PBS at RT for 2 min for nuclear staining. After washing three times, the ovarian cells were observed under a fluorescence microscope (DM2500, Leica, Wetzlar, Germany) at 200× magnification.

### 2.3. RNA extraction and RNASeq analysis

Total RNAs were extracted from the whole bodies of female *F. occidentalis* at different ages (0, 36, and 60 h after adult emergence). Three independent samples were used for three

replications at each age. Each sample consisted of 50 females. RNA extraction was performed using Trizol reagent (Invitrogen, Carlsbad, CA, USA) according to the manufacturer's instructions. An RNA library was generated using TruSeq Stranded Total RNA with Ribo-Zero H/M/R_Gold (Illumina, San Diego, CA, USA). RNA sequencing was performed on the NovaSeq 6000 platform (Illumina) from Macrogen (Seoul, Korea). The RNA sequence was trimmed using CLC Workbench (QIAgen, Hilden, Germany). To calculate relative transcript accumulation, reads per kilobase per million (RPKM) mappable reads of the *F. occidentalis* genome (GenBank accession number: GCF_000697945.2) were estimated using CLC Workbench based on a template of Focc_2.1 version with a trimmed sequence of more than 50 bp.

## 2.4. Bioinformatics

DEGs were selected based on a fold change of $\geq 2.0$ and a *P*-value of $< 0.05$ with three biological replicates by comparing RPKM values at 36 h or 60 h after adult emergence (AAE) to those at 0 h AAE. KEGG pathway analysis was performed to test the statistical enrichment of DEGs using the KEGG mapper (https://www.kegg.jp/kegg/) by converting the National Center for Biotechnology Information (NCBI) Gene ID to a KEGG ID through the convert ID tool of KEGG mapper.

## 2.5. RT-qPCR

After RNA extraction, RNA was resuspended in nuclease-free water and quantified using a spectrophotometer (NanoDrop, Thermo Fisher Scientific, Wilmington, DE, USA). RNA (500 ng) was used for cDNA synthesis with RT PreMix (Intron Biotechnology, Seoul, Korea) containing oligo dT primers according to the manufacturer's instructions. All gene expression levels in this study were determined using a real-time polymerase chain reaction (PCR) machine (Step One Plus Real-Time PCR System, Applied Biosystem, Singapore) under the guidelines of [9]. Real-time PCR was conducted in a reaction volume of 20 μL containing 10 μL of Power SYBR Green PCR Master Mix (Thermo Scientific Korea), 3 μL of cDNA template (200 ng), and each 1 μL (10 pmol) of forward and reverse primers (S1 Table). After initial heat treatment at 95°C for 2 min, qPCR was performed with 40 cycles of denaturation at 95°C for 30 sec, annealing at 53 ~ 55°C for 30 sec, and extension at 72°C for 30 sec. The expression level of elongation factor-1 (*EF-1*, S1 Table) was used as a reference to normalize the target gene expression levels under different treatments. Quantitative analysis was performed using the comparative CT ($2^{-\Delta\Delta CT}$) method [10]. All experiments were independently replicated three times.

## 2.6. Statistical analysis

All the continuous variable data were subjected to a one-way analysis of variance using PROC GLM in the SAS program [11]. Means were compared with Duncan's multiple range test (DMRT) at type I error = 0.05.

# 3. Results

## 3.1. Ovarian development in *F. occidentalis*

Dissection of 3-day-old female adults showed that a pair of ovaries contained eight ovarioles (Fig 1A). The four ovarioles in each ovary were joined to a lateral oviduct and the two lateral oviducts were combined to the common oviduct. Each ovariole contained a string of follicles and was divided into a germarium and a vitellarium depending upon the presence of matured oocytes (Fig 1B). In the germarium, small cells were closely attached and formed a germ cell

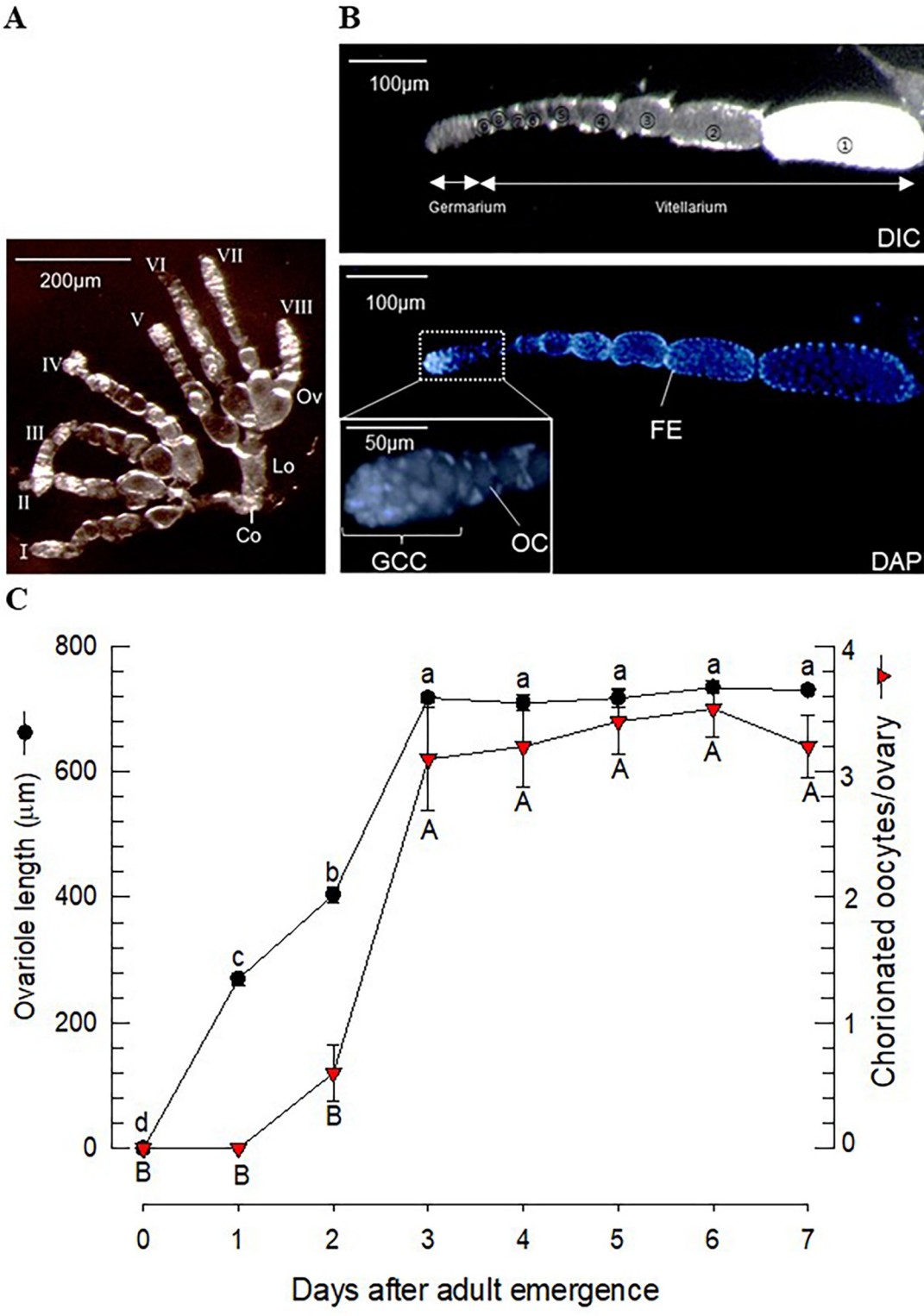

**Fig 1. Ovarian development of *F. occidentalis.*** (A) Internal reproductive organs of 2-day-old females include a pair of ovaries ('Ov'), 8 ovarioles ('I-VIII'), a lateral oviduct ('Lo'), and a common oviduct ('Co'). (B) Ovariole structure divided into the germarium and the vitellarium. A total of 9 follicles are numbered from proximally to distally in the differential interference contrast ('DIC') picture. The white follicle represents chorionated oocytes. In the DAPI picture, each oocyte in the follicle is surrounded by follicular epithelium ('FE'). In the germarium, the germ cell cluster ('GCC') is located near to the oocyte ('OC'). (C)

Ovarian development with female age. The ovariole length represents the germarium and vitellarium. Chorionated oocytes were counted per ovary. Each measurement used individual thrips and was replicated 5 times.

cluster. The vitellarium contained nine follicles, each of which was composed of oocytes and follicular epithelium without any nurse cells. The terminal follicles were usually chorionated.

Ovarian development occurred after adult emergence (Fig 1C). The ovariole length began to extend just after adult emergence and reached a maximal size three days after emergence. Chorionated oocytes were visible after two days in some ovarioles. After three days, all the ovarioles had terminal chorionated oocytes, but some of the ovarioles lost them due to oviposition. This ovarian development pattern allowed us to determine three stages: early at 0 h, mid at 36 h, and late at 60 h after adult emergence.

## 3.2. Changes in the transcriptomes of female *F. occidentalis* adults during ovarian development

Total RNA was sequenced at three ovarian developmental stages of *F. occidentalis*. Each of the nine samples (= 3 stages × 3 replications) was sequenced from 10~13 Gb (Table 1). After trimming, 102~131 million reads in each sample were used to map to the *F. occidentalis* genome. With 71~82% mapping rates, the reads in each sample were matched to 14,042~14,521 genes among 16,859 predicted *F. occidentalis* genes. All the nine transcriptomes were deposited to GenBank with accession numbers of PRJNA833754.

When the transcriptomes of the three developmental stages were compared, they shared more than 95% (= 14,370/15,055) of the transcripts (Fig 2A). Forty-nine unique genes expressed at 36 h AAE were classified according to structure, gene regulation, and cell cycle along with several uncharacterized genes (S2 Table). However, their expression levels were extremely low at 0.0007~0.0468 RPKM. Forty-seven unique genes expressed at 60 h AAE were classified according to structure, protein-processing, and gene regulation along with several uncharacterized genes (S3 Table). Their expression levels were relatively high at 1.0016~1.6941 RPKM. However, this unique gene analysis did not identify genes apparently associated with oogenesis.

**Table 1. Sequencing summary of *F. occidentalis* transcripts at different female ages.**

| Age[1] | N[2] | Total sequences[3] (bp) | Trimmed reads[4] (bp) | Mapping[5] (%) | Matched genes[6] |
|---|---|---|---|---|---|
| **0 h** | 1 | 10,320,812,260 | 102,185,476 | 71.35 | 14,298 |
| | 2 | 11,065,530,710 | 109,558,978 | 80.16 | 14,467 |
| | 3 | 13,142,766,804 | 130,125,490 | 78.48 | 14,521 |
| **36 h** | 1 | 13,287,362,040 | 131,557,128 | 79.03 | 14,042 |
| | 2 | 12,494,164,196 | 123,703,708 | 82.58 | 14,154 |
| | 3 | 12,040,868,520 | 126,733,690 | 80.40 | 14,145 |
| **60 h** | 1 | 12,800,190,964 | 131,932,260 | 73.82 | 14,266 |
| | 2 | 13,325,257,038 | 126,848,436 | 81.45 | 14,211 |
| | 3 | 12,811,780,916 | 126,848,436 | 73.88 | 14,119 |

[1]Age represents the time (h) after adult emergence of females.

[2] N represents the number of replications. Each replication used 50 females for RNA extraction.

[3]Sequenced by the NovaSeq platform (Illumina, San Diego, CA, USA).

[4]Trimmed by CLC Workbench (QIAgen, Hilden, Germany).

[5]Mapping to the *F. occidentalis* genome (GenBank accession number: GCF_000697945.2).

[6]Total number of annotated genes was 16,859.

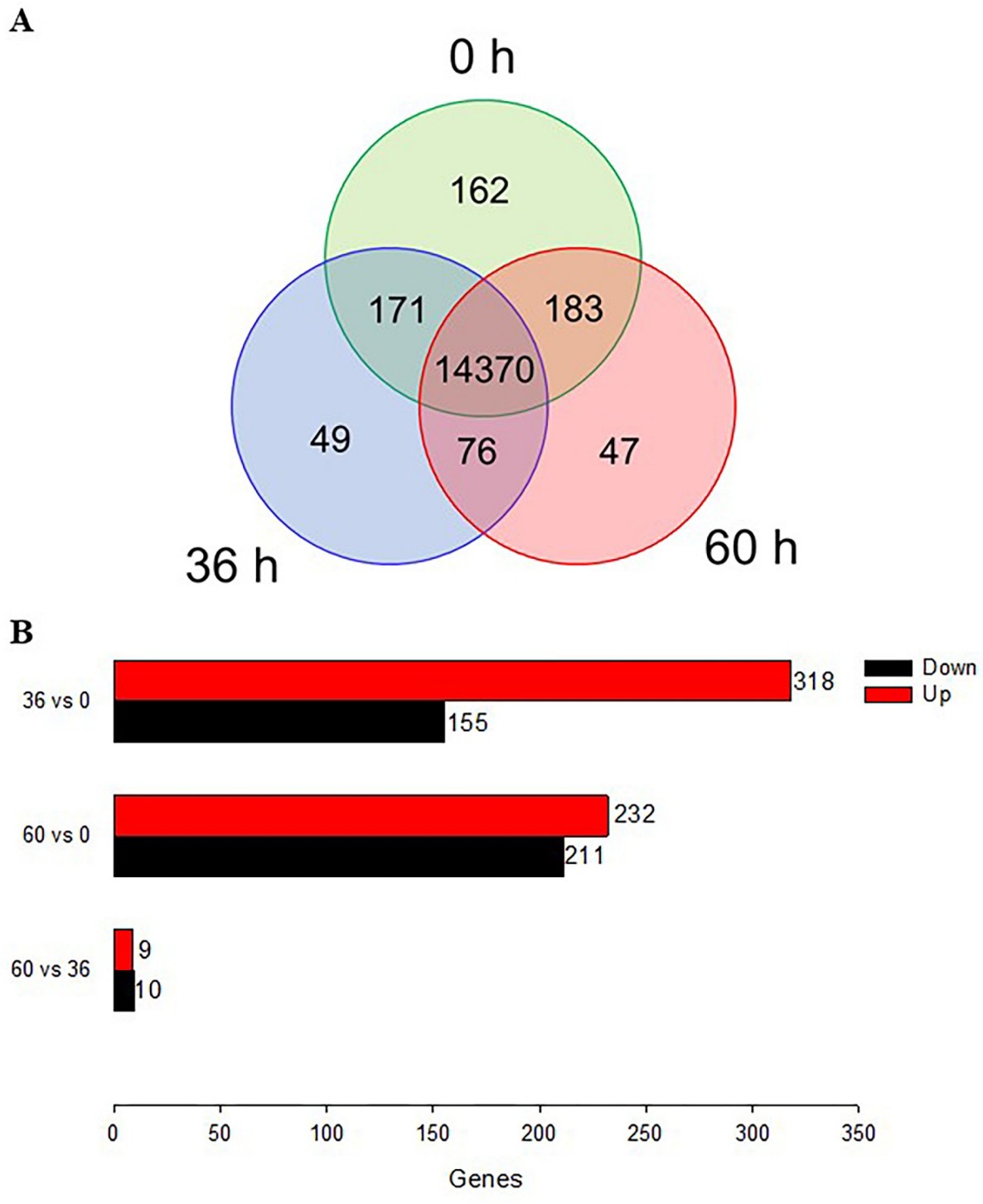

**Fig 2. DEG analysis of different ages of *F. occidentalis* females.** The assessment used genes mapped with RNASeq reads (see Table 1). (A) Venn diagram of total transcripts at 0 h, 36 h, and 60 h after emergence. (B) DEG analysis between two different developmental stages. The threshold was > 2-fold changes in RPKM.

To identify *F. occidentalis* genes associated with ovarian development, a differentially expressed gene (DEG) analysis was performed (Fig 2B). At 36 h AAE, 473 transcripts showed more than 2-fold increases in gene expression levels compared to those at 0 h AAE. At 60 h AAE, 443 transcripts showed more than 2-fold increases in gene expression levels compared to those at 0 h AAE. Only 19 transcripts showed changes between 36 h and 60 h AAE.

To characterize the DEGs at the mid (36 h AAE) and late (60 h AAE) ovarian developmental stages, their gene functions were predicted using KEGG analysis (Fig 3). Both DEGs were assigned to 52 KEGG functional categories but they did not exactly overlap. The DEGs of each developmental stage were assigned to 49 KEGG categories with three different missing categories. DEGs at 36 h AAE did not include the three categories of #21 (FoxO signaling pathway), #42 (protein export), and #50 (N-glycan biosynthesis), whereas the DEGs at 60 h AAE did not include the three categories of #17 (fatty acid biosynthesis), #30 (lysine degradation), and #39 (pentose and glucuronate interconversion). Among 46 common KEGG categories, most DEGs were classified into the metabolic pathway category (#33) in both developmental stages. The other major (> 10 DEGs) categories, biosynthesis of cofactors (#8) and lysosome (#31), were common in both developmental stages. However, the 36 h AAE samples had more DEGs in the inositol phosphate metabolism category (#28) than the 60 h AAE samples. In contrast, the glycan biosynthesis (#35) and purine metabolism (#44) categories had more DEGs at 60 h AAE than at 36 h AAE. The KEGG analysis suggested the upregulation of metabolic pathways associated with nucleic acid biosynthesis during ovarian development.

## 3.3. Expression profiles of egg proteins during ovarian development

To identify the specific genes associated with oogenesis in *F. occidentalis*, we selected genes that were highly expressed more than 8-fold at 36 h AAE or 60 h AAE compared to expression levels at 0 h AAE (Table 2). The selected 99 genes were subdivided into structure, protein processing, lipid metabolism, gene regulation, and others. The structural protein category included typical egg proteins such as vitellogenin, chorion protein, mucin, and yellow melanization protein. In contrast, the highly suppressed genes at these stages included 37 genes (S4 Table). Especially, larval cuticular protein genes were included in the suppressed gene category.

The expression patterns of representative egg proteins during adult development were further analyzed (Fig 4). As expected, RNASeq analysis found that *vitellogenin*, *chorion protein*, *mucin*, and *yellow* genes were highly expressed in the mid and late ovarian developmental stages (Fig 4A). However, there was little or no difference in the expression levels between the two developmental stages (36 h AAE and 60 h AAE). These transcript level profiles were further assessed by RT-qPCR with additional development stages (Fig 4B). The expression levels measured by RT-qPCR were mostly consistent with the expression profiles measured by RNA-Seq. However, RT-qPCR analysis indicated that the *mucin* and *yellow* genes were induced earlier than 36 h AAE. It also showed their expression patterns in late ovarian development after 60 h AAE, in which *vitellogenin* and *mucin* maintained the induced levels, whereas the expression levels of *chorion protein* and *yellow* significantly decreased.

## 3.4. Expression profiles of genes associated with endocrine signals during ovarian development

Juvenile hormone (JH), ecdysteroid, and insulin-like peptide (ILP) are well-known endocrine mediators during insect reproduction [12]. Genes associated with these endocrine signals were selected from the transcriptomes (Table 3). JH acid methyltransferase (JHAMT), JH esterase/ JH epoxide hydrolase, and Met involved in JH synthesis, JH degradation, and the JH receptor,

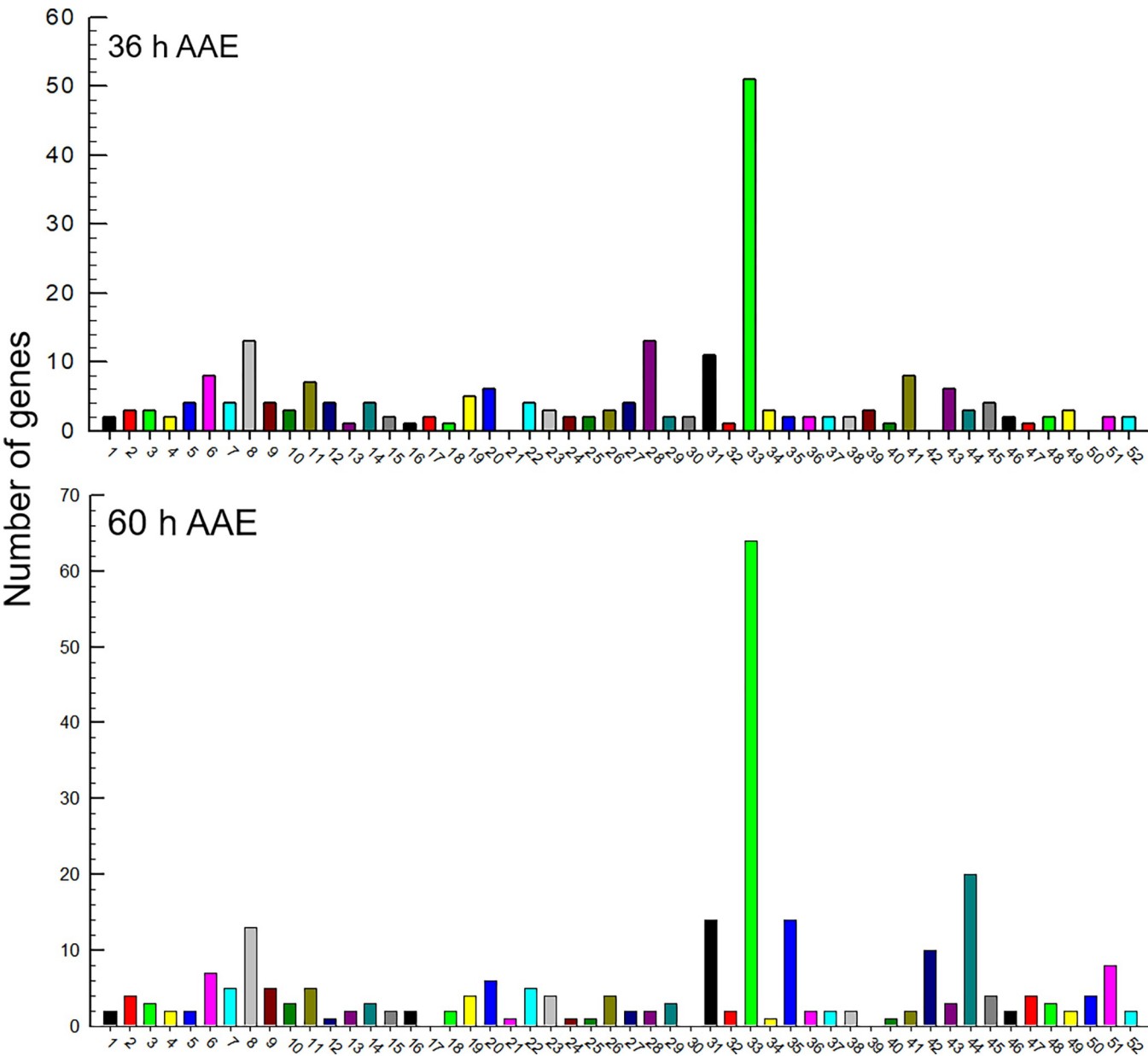

**Fig 3. KEGG analysis of the DEGs selected from Fig 2B.** The metabolic pathways in the KEGG database were mapped with the DEGs. 36 h AAE represents DEGs between transcripts at 0 h and 36 h after adult emergence and 60 h AAE represents DEGs between transcripts at 0 h and 60 h after adult emergence. KEGG categories include alanine, aspartate and glutamate metabolism (1), amino sugar and nucleotide sugar metabolism (2), apoptosis (3), arachidonic acid metabolism (4), ascorbate and aldarate metabolism (5), autophagy (6), biosynthesis of amino acids (7), biosynthesis of cofactors (8), biosynthesis of nucleotide sugars (9), biosynthesis of unsaturated fatty acids (10), carbon metabolism (11), citrate cycle (12), cysteine and methionine metabolism (13), cytochrome P450 (14), ECM-receptor interaction (15), endocytosis (16), fatty acid biosynthesis (17), fatty acid elongation (18), fatty acid metabolism (19),folate biosynthesis (20), foxO signaling pathway (21), fructose and mannose metabolism (22), glutathione metabolism (23), glycerolipid metabolism (24), glycerophospholipid metabolism (25), glycine, serine and threonine metabolism (26), glycolysis/gluconeogenesis (27), inositol phosphate metabolism (28), longevity regulating pathway (29), lysine degradation (30), lysosome (31), MAPK signaling pathway (32), metabolic pathways (33), mitophagy (34), glycan biosynthesis (35), nicotinate and nicotinamide metabolism (36), nucleotide metabolism (37), oxidative phosphorylation (38), pentose and glucoronate interconversions (39), pentose phosphate pathway (40), phagosome (41), protein export (42), protein processing in endoplasmic reticulum (43), purine metabolism (44), pyruvate metabolism (45), starch and sucrose metabolism (46), steroid biosynthesis (47), thiamine metabolism (48), Toll and Imd signaling pathway (49), various types of N-glycan biosynthesis (50), vitamin B6 metabolism (51), and Wnt signaling pathway (52).

**Table 2. Highly (> 8-fold) expressed genes at mid (36 h after adult emergence) and late (60 h) ovarian development stages compared to expression levels at the early (0 h) developmental stage in female _F. occidentalis_ adults.**

| Category (99) | NCBI Gene ID | GenBank accession | Annotation | 36 h | | 60 h | |
|---|---|---|---|---|---|---|---|
| | | | | RPKM | Log$_2$ Fc | RPKM | Log$_2$ Fc |
| Structure (18) | LOC113214498 | XM_026433864.1 | vitellogenin | 2.69 | 4.13 | 2.58 | 3.77 |
| | LOC113214411 | XM_026433762.1 | vitellogenin-2 | 2.60 | 4.44 | 2.58 | 4.38 |
| | LOC113202497 | XM_026416747.1 | vitellogenin-1 | 2.60 | 3.35 | 2.64 | 3.51 |
| | LOC113204471 | XM_026419639.1 | vitellogenin-1 | 2.75 | 4.05 | 2.73 | 4.01 |
| | LOC113212925 | XM_026431801.1 | vitellogenin-1 | 1.68 | 3.71 | 1.63 | 3.54 |
| | LOC113205436 | XM_026421068.1 | flocculation protein | 1.21 | 3.42 | 1.07 | 2.94 |
| | LOC113214189 | XM_026433479.1 | endoglucanase-5 | 0.95 | 2.53 | 1.17 | 3.27 |
| | LOC113211573 | XM_026429977.1 | chorion class A protein | 2.70 | 5.12 | 2.56 | 4.67 |
| | LOC113202531 | XM_026416797.1 | myosin heavy chain | 1.70 | 4.05 | 1.54 | 3.50 |
| | LOC113213533 | XM_026432640.1 | mucin-5AC | 1.87 | 4.25 | 1.82 | 4.06 |
| | LOC113217829 | XM_026437896.1 | filaggrin | 1.04 | 3.33 | 0.95 | 3.01 |
| | LOC113206591 | XM_026422740.1 | filaggrin-2 | 1.19 | 3.02 | 1.15 | 2.89 |
| | LOC113217875 | XM_026437953.1 | keratin-associated protein | 1.67 | 4.55 | 1.58 | 4.27 |
| | LOC113217915 | XM_026438006.1 | keratin-associated protein | 1.64 | 4.84 | 1.56 | 4.56 |
| | LOC113218104 | XM_026438307.1 | protein yellow | 2.70 | 5.29 | 2.62 | 5.00 |
| | LOC113206471 | XM_026422568.1 | nacrein | 2.34 | 3.94 | 2.41 | 4.18 |
| | LOC113214093 | XM_026433360.1 | sperm acrosome-associated protein 5 | 2.36 | 5.65 | 2.50 | 6.12 |
| | LOC113217267 | XM_026437086.1 | venom allergen | 1.96 | 5.17 | 2.02 | 5.39 |
| Protein processing (23) | LOC113216008 | XM_026435682.1 | transmembrane protein | 2.16 | 5.40 | 2.05 | 5.03 |
| | LOC113214127 | XM_026433402.1 | PE-PGRS family protein | 2.08 | 3.50 | 1.83 | 2.67 |
| | LOC113209730 | XM_026427399.1 | trypsin | 1.69 | 3.73 | 1.39 | 2.72 |
| | LOC113204180 | XM_026419243.1 | trypsin | 1.95 | 3.66 | 1.77 | 3.07 |
| | LOC113203267 | XM_026417851.1 | cathepsin L1 | 3.28 | 3.01 | 3.17 | 2.66 |
| | LOC113205904 | XM_026421697.1 | cathepsin L1 | 1.57 | 2.81 | 1.71 | 3.28 |
| | LOC113212259 | XM_026430877.1 | carboxypeptidase B | 1.08 | 3.16 | 1.04 | 3.03 |
| | LOC113202896 | XM_026417339.1 | transmembrane protease serine 9 | 1.75 | 3.26 | 1.53 | 2.54 |
| | LOC113206464 | XM_026422562.1 | transmembrane protease serine 9 | 1.97 | 4.45 | 1.89 | 4.20 |
| | LOC113214490 | XM_026433856.1 | probable pectin lyase D | 1.93 | 4.93 | 2.09 | 5.45 |
| | LOC113207878 | XM_026424617.1 | probable pectin lyase B | 2.13 | 3.87 | 2.02 | 3.50 |
| | LOC113216785 | XM_026436611.1 | probable pectin lyase B | 2.93 | 5.59 | 3.11 | 6.17 |
| | LOC113216787 | XM_026436612.1 | probable pectin lyase B | 1.93 | 4.76 | 1.99 | 4.95 |
| | LOC113211651 | XM_026430098.1 | pectin lyase | 1.27 | 3.11 | 1.11 | 2.57 |
| | LOC113203564 | XM_026418330.1 | pectin lyase | 1.16 | 2.79 | 1.24 | 3.05 |
| | LOC113215606 | XM_026435246.1 | lysozyme C milk isozyme | 1.28 | 3.61 | 1.43 | 4.11 |
| | LOC113217852 | XM_026437921.1 | polyhomeotic-proximal chromatin protein | 1.30 | 3.46 | 1.19 | 3.09 |
| | LOC113208437 | XM_026425417.1 | transcriptional regulatory protein AlgP | 2.03 | 3.32 | 2.19 | 3.85 |
| | LOC113217886 | XM_026437969.1 | cyclin-dependent kinase inhibitor | 1.38 | 4.24 | 1.41 | 4.36 |
| | LOC113210725 | XM_026428834.1 | PE-PGRS family protein | 2.42 | 3.86 | 2.14 | 2.93 |
| | LOC113212288 | XM_026430917.1 | protein rtoA | 0.80 | 2.05 | 1.24 | 3.52 |
| | LOC113207332 | XM_026423852.1 | hornerin | 1.96 | 3.14 | 1.98 | 3.21 |
| | LOC113206614 | XM_026422771.1 | hornerin | 2.12 | 3.84 | 2.10 | 3.75 |
| Lipid metabolism (6) | LOC113208001 | XM_026424805.1 | clavesin-1 | 1.31 | 3.63 | 1.17 | 3.15 |
| | LOC113213966 | XM_026433193.1 | pancreatic triacylglycerol lipase | 1.33 | 3.17 | 1.33 | 3.16 |
| | LOC113215706 | XM_026435368.1 | low-density lipoprotein receptor | 1.20 | 3.38 | 1.06 | 2.94 |
| | LOC113214129 | XM_026433404.1 | lipase member K | 1.02 | 3.03 | 1.05 | 3.14 |
| | LOC113202163 | XM_026416266.1 | acyl-CoA Delta (11) desaturase | 1.80 | 3.82 | 1.45 | 2.66 |
| | LOC113202156 | XM_026416257.1 | phospholipase A1 | 1.73 | 3.00 | 1.60 | 2.58 |
| Gene regulation (3) | LOC113211421 | XM_026429785.1 | serine-rich adhesin for platelets | 1.44 | 4.15 | 1.41 | 4.06 |
| | LOC113217849 | XM_026437920.1 | serine-rich adhesin for platelets | 1.58 | 4.98 | 1.50 | 4.73 |
| | LOC113210704 | XM_026428809.1 | regucalcin | 1.50 | 3.58 | 1.48 | 3.51 |

_(Continued)_

**Table 2.** (Continued)

| Category (99) | NCBI Gene ID | GenBank accession | Annotation | 36 h | | 60 h | |
|---|---|---|---|---|---|---|---|
| | | | | RPKM | Log$_2$ Fc | RPKM | Log$_2$ Fc |
| Others (49) | LOC113204174 | XM_026419235.1 | histidine-rich glycoprotein | 2.05 | 4.91 | 1.86 | 4.27 |
| | LOC113216845 | XM_026436674.1 | histidine-rich glycoprotein | 3.02 | 4.16 | 2.98 | 4.04 |
| | LOC113217182 | XM_026437013.1 | non-classical arabinogalactan protein | 1.11 | 3.55 | 0.97 | 3.06 |
| | LOC113202260 | XM_026416395.1 | neurofilament medium polypeptide | 1.58 | 4.19 | 1.42 | 3.65 |
| | LOC113202298 | XM_026416451.1 | proline-rich protein 2 | 2.10 | 3.76 | 2.03 | 3.55 |
| | LOC113217885 | XM_026437968.1 | uncharacterized | 2.53 | 4.89 | 2.45 | 4.62 |
| | LOC113217883 | XM_026437966.1 | uncharacterized | 1.33 | 4.33 | 1.25 | 4.07 |
| | LOC113217830 | XM_026437897.1 | uncharacterized | 1.53 | 4.33 | 1.49 | 4.18 |
| | LOC113217828 | XM_026437895.1 | uncharacterized | 1.08 | 3.32 | 0.97 | 2.97 |
| | LOC113217577 | XM_026437533.1 | uncharacterized | 1.60 | 4.62 | 1.93 | 5.71 |
| | LOC113217326 | XM_026437181.1 | uncharacterized | 1.67 | 3.66 | 1.69 | 3.74 |
| | LOC113216895 | XM_026436728.1 | uncharacterized | 1.13 | 2.47 | 1.30 | 3.02 |
| | LOC113216056 | XM_026435756.1 | uncharacterized | 2.67 | 4.35 | 2.69 | 4.42 |
| | LOC113216047 | XM_026435745.1 | uncharacterized | 1.47 | 3.39 | 1.21 | 2.53 |
| | LOC113214206 | XM_026433496.1 | uncharacterized | 1.98 | 3.53 | 1.92 | 3.32 |
| | LOC113213753 | XM_026432913.1 | uncharacterized | 1.79 | 3.62 | 1.66 | 3.21 |
| | LOC113212826 | XM_026431653.1 | uncharacterized | 1.70 | 4.11 | 1.53 | 3.56 |
| | LOC113212825 | XM_026431652.1 | uncharacterized | 1.50 | 3.68 | 1.34 | 3.15 |
| | LOC113212293 | XM_026430924.1 | uncharacterized | 0.68 | 1.85 | 1.07 | 3.14 |
| | LOC113212290 | XM_026430919.1 | uncharacterized | 0.87 | 2.52 | 1.28 | 3.89 |
| | LOC113212282 | XM_026430908.1 | uncharacterized | 2.63 | 3.06 | 2.60 | 2.98 |
| | LOC113211882 | XM_026430401.1 | uncharacterized | 1.77 | 4.33 | 1.58 | 3.72 |
| | LOC113211714 | XM_026430181.1 | uncharacterized | 2.37 | 4.68 | 2.31 | 4.48 |
| | LOC113211400 | XM_026429765.1 | uncharacterized | 1.54 | 4.46 | 1.93 | 5.75 |
| | LOC113211399 | XM_026429764.1 | uncharacterized | 1.69 | 3.72 | 2.00 | 4.74 |
| | LOC113211078 | XM_026429332.1 | uncharacterized | 2.19 | 4.85 | 2.24 | 4.99 |
| | LOC113210730 | XM_026428840.1 | uncharacterized | 1.99 | 3.31 | 1.85 | 2.87 |
| | LOC113210583 | XM_026428652.1 | uncharacterized | 1.74 | 3.00 | 1.70 | 2.86 |
| | LOC113210342 | XM_026428298.1 | uncharacterized | 1.79 | 4.37 | 1.97 | 4.99 |
| | LOC113208836 | XM_026426044.1 | uncharacterized | 1.39 | 4.26 | 1.29 | 3.93 |
| | LOC113208574 | XM_026425626.1 | uncharacterized | 1.32 | 3.16 | 1.28 | 3.04 |
| | LOC113208471 | XM_026425481.1 | uncharacterized | 1.62 | 3.41 | 1.54 | 3.13 |
| | LOC113208012 | XM_026424840.1 | uncharacterized | 2.21 | 3.98 | 1.92 | 3.01 |
| | LOC113206973 | XM_026423287.1 | uncharacterized | 2.18 | 3.50 | 1.90 | 2.57 |
| | LOC113206617 | XM_026422774.1 | uncharacterized | 1.49 | 3.41 | 1.28 | 2.70 |
| | LOC113205710 | XM_026421427.1 | uncharacterized | 2.21 | 5.77 | 2.32 | 6.12 |
| | LOC113205709 | XM_026421426.1 | uncharacterized | 2.13 | 5.75 | 2.23 | 6.09 |
| | LOC113205708 | XM_026421425.1 | uncharacterized | 2.56 | 5.69 | 2.73 | 6.24 |
| | LOC113205435 | XM_026421067.1 | uncharacterized | 1.49 | 3.43 | 1.28 | 2.72 |
| | LOC113205325 | XM_026420899.1 | uncharacterized | 2.30 | 4.79 | 2.29 | 4.77 |
| | LOC113205123 | XM_026420596.1 | uncharacterized | 1.29 | 3.62 | 0.84 | 2.11 |
| | LOC113204714 | XM_026419980.1 | uncharacterized | 1.79 | 3.52 | 1.67 | 3.14 |
| | LOC113204504 | XM_026419693.1 | uncharacterized | 1.43 | 3.32 | 1.45 | 3.39 |
| | LOC113204057 | XM_026419056.1 | uncharacterized | 2.05 | 3.55 | 2.03 | 3.51 |
| | LOC113203620 | XM_026418390.1 | uncharacterized | 1.57 | 4.42 | 1.88 | 5.46 |
| | LOC113203560 | XM_026418318.1 | uncharacterized | 1.52 | 3.65 | 1.52 | 3.66 |
| | LOC113203559 | XM_026418317.1 | uncharacterized | 1.33 | 3.18 | 1.25 | 2.89 |
| | LOC113203548 | XM_026418307.1 | uncharacterized | 2.25 | 4.83 | 2.29 | 4.96 |
| | LOC113202588 | XM_026416887.1 | uncharacterized | 2.39 | 4.90 | 2.25 | 4.43 |

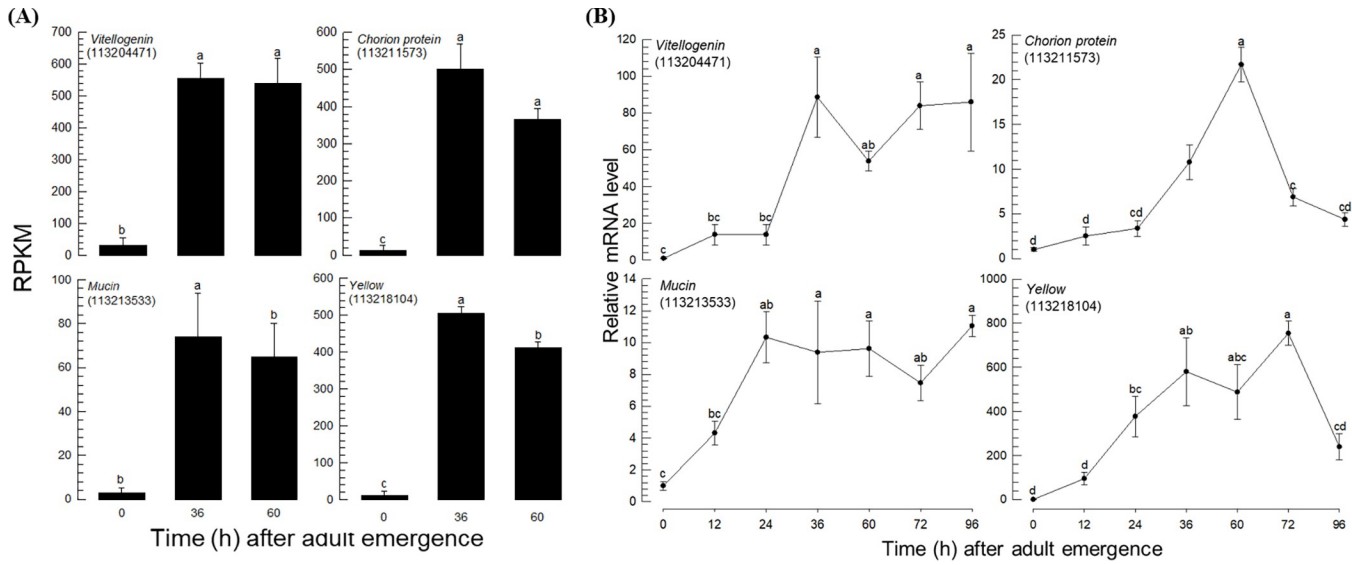

**Fig 4. Expression analysis of the selected egg proteins during *F. occidentalis* ovarian development.** The NCBI Gene ID is in each parenthesis. (A) RNASeq analysis. (B) RT-qPCR analysis. Each measurement was replicated 3 times. The figures in parentheses indicate LOCs, which are the NCBI gene IDs. Different letters above the standard deviation bars represent significant differences among the means at type I error = 0.05 (DMRT test)].

respectively, were identified. The RNASeq analysis showed that JHAMT was highly upregulated at 36 h AAE, whereas Met expression levels did not change during adult development (Fig 5A). As ecdysteroid signaling genes, *Shade* and *EcR* were found in the adult transcriptomes. The expression levels of *Shade* were upregulated at mid and late ovarian development, during which *EcR* expression was slightly decreased. Insulin-like peptides and receptors were included in the adult transcriptomes. *ILP* was highly upregulated at 36 h AAE, whereas *InR* expression levels decreased during adult development. The transcriptome profiles related to endocrine signals were further supported by RT-qPCR analysis (Fig 5B). The RT-qPCR analysis assessed at several time points after adult emergence showed the upregulation of these endocrine signals during ovarian development. Especially, JH and insulin signals were more upregulated at 36 h AAE than at 60 h AAE, whereas the ecdysteroid signal was more upregulated at the late developmental stage.

## 4. Discussion

Various reproductive modes are observed in Thysanoptera. In *F. occidentalis*, a female produces progeny in bisexual or asexual arrhenotokous reproduction [13]. In arrhenotokous reproduction in the situation of few males (e.g., overwintering population), virgin females produce only male offspring [6]. When their sons are sexually mature, the females undergo bisexual reproduction with their sons and produce female-biased offspring. All thrips including *F. occidentalis* exhibit oviparous reproduction, in which females in either the asexual or bisexual mode undergo oogenesis in their ovaries. However, little is known about ovarian development in *F. occidentalis*. The current study analyzed the internal reproductive organ structure of *F. occidentalis* to assess the developing oocytes. Based on the temporal developmental pattern, RNASeq analysis was performed at early, mid, and late stages to determine the specific genes associated with ovarian development.

   The ovaries and associated internal reproductive organs of *F. occidentalis* were observed. Eight ovarioles from a pair of ovaries contained follicles, the end of which contained

**Table 3. DEGs associated with endocrine signals in female *F. occidentalis* at mid (36 h after adult emergence) and late ovarian development stages compared to the early (0 h) stage.**

| Category | NCBI Gene ID | GenBank Accession | Annotation | 36 h | | 60 h | |
|---|---|---|---|---|---|---|---|
| | | | | RPKM | Log₂Fc | RPKM | Log₂Fc |
| **Juvenile hromone (JH)** | LOC113205672 | XM_026421382.1 | JH esterase | 2.78 | 0.11 | 2.08 | -0.31 |
| | LOC113206119 | XM_026422046.1 | JH epoxide hydrolase | 48.44 | 2.32 | 37.15 | 1.94 |
| | LOC113206791 | XM_026423048.1 | JH esterase | 6.75 | 1.66 | 5.04 | 1.24 |
| | LOC113207106 | XM_026423497.1 | JH esterase | 0.61 | -1.74 | 0.48 | -2.09 |
| | LOC113207121 | XM_026423514.1 | JH esterase | 0.17 | -0.38 | 0.11 | -1.06 |
| | LOC113208804 | XM_026425987.1 | JH esterase | 0.92 | -1.16 | 1.17 | -0.81 |
| | LOC113209084 | XM_026426427.1 | JH esterase | 0.77 | 1.18 | 0.59 | 0.81 |
| | LOC113211838 | XM_026430351.1 | JH esterase | 8.02 | 4.27 | 8.17 | 4.30 |
| | LOC113215323 | XM_026434935.1 | JH-suppressible protein | 0.72 | -3.37 | 0.23 | -5.01 |
| | LOC113202122 | XM_026416212.1 | JH esterase | 3.67 | -0.28 | 3.48 | -0.36 |
| | LOC113202626 | XM_026416948.1 | JH acid methyltransferase | 1.17 | 0.61 | 0.57 | -0.43 |
| | LOC113202308 | XM_026416465.1 | JH esterase | 0.95 | -0.30 | 0.77 | -0.60 |
| | LOC113217970 | XM_026416465.2 | Methoprene tolerance | 2.99 | 0.10 | 2.60 | -0.10 |
| **Ecdysteroid** | LOC113205624 | XM_026421313.1 | Shade | 3.32 | 0.51 | 2.96 | 0.35 |
| | LOC113207454 | XM_026424033.1 | ecdysone receptor | 0.21 | -1.80 | 0.18 | -2.04 |
| | LOC113211564 | XM_026429966.1 | ecdysone receptor | 29.17 | -0.11 | 23.39 | -0.42 |
| | LOC113211835 | XM_026430346.1 | zinc finger protein on ecdysone | 4.31 | 0.22 | 4.09 | 0.14 |
| | LOC113211939 | XM_026430486.1 | ecdysone-induced protein | 0.77 | -1.43 | 0.42 | -2.33 |
| | LOC113214771 | XM_026434236.1 | protein ecdysoneless | 6.68 | 0.23 | 6.47 | 0.18 |
| | LOC113216508 | XM_026436255.1 | protein ecdysoneless | 1.64 | 0.02 | 1.64 | 0.02 |
| | LOC113216945 | XM_026436781.1 | ecdysone-induced protein 74EF | 1.72 | -1.37 | 1.74 | -1.35 |
| **Insulin** | LOC113202214 | XM_026416323.1 | IGF1 receptor | 0.10 | -1.14 | 0.10 | -1.17 |
| | LOC113206117 | XM_026422043.1 | insulin-like receptor | 5.22 | -0.27 | 4.67 | -0.43 |
| | LOC113207224 | XM_026423730.1 | insulin-like peptide | 4.67 | 1.25 | 2.36 | 0.27 |
| | LOC113209425 | XM_026426931.1 | IGF2-BP | 17.36 | 0.50 | 13.70 | 0.16 |
| | LOC113210134 | XM_026427972.1 | insulin-degrading enzyme | 8.92 | 0.03 | 8.04 | -0.12 |
| | LOC113211136 | XM_026429429.1 | IGF-BP complex acid labile subunit | 68.21 | 0.31 | 48.47 | -0.19 |
| | LOC113211621 | XM_026430058.1 | IGF-BP complex acid labile subunit | 2.53 | 0.47 | 1.85 | 0.02 |
| | LOC113211864 | XM_026430384.1 | IGF-BP complex acid labile subunit | 9.49 | -0.57 | 6.76 | -1.06 |
| | LOC113212596 | XM_026431366.1 | IGF-BP complex acid labile subunit | 2.06 | -0.49 | 1.56 | -0.89 |
| | LOC113212620 | XM_026431414.1 | IGF-BP complex acid labile subunit | 0.69 | -0.70 | 0.49 | -1.21 |
| | LOC113215479 | XM_026435105.1 | IGF-BP7 | 0.09 | -0.75 | 0.06 | -1.34 |

chorionated oocytes. Each follicle was composed of oocytes and follicular epithelium. The distal germarium contained a germ cell cluster, presumably consisting of interconnected nurse cells and oocytes. The ovarioles of insects are categorized into panoistic and meroistic types, in which the latter type is subdivided into polytrophic and telotrophic groups [14]. The panoistic type is considered to be the most ancestral because of its deficiency in transforming oogonia into nurse cells [15]. The polytrophic meroistic ovary has evolved from the panoistic ovary through the differentiation of nurse cells, and finally, the telotrophic meroistic type is derived from the polytrophic meroistic type by the restriction of nurse cells to the germarium. A deviation from the typical ovariole types is observed in Thysanoptera, in which the germ cell cluster is formed as seen in the ovary of the terebrantian thrip, *Purthenothrips drucenae* [16], suggesting that the panoistic follicles resulted from the secondary loss of nurse cells from the germ cell cluster. Stys et al. [14] called this type of ovary "neo-panoistic." Thus, the thysanopteran ovary

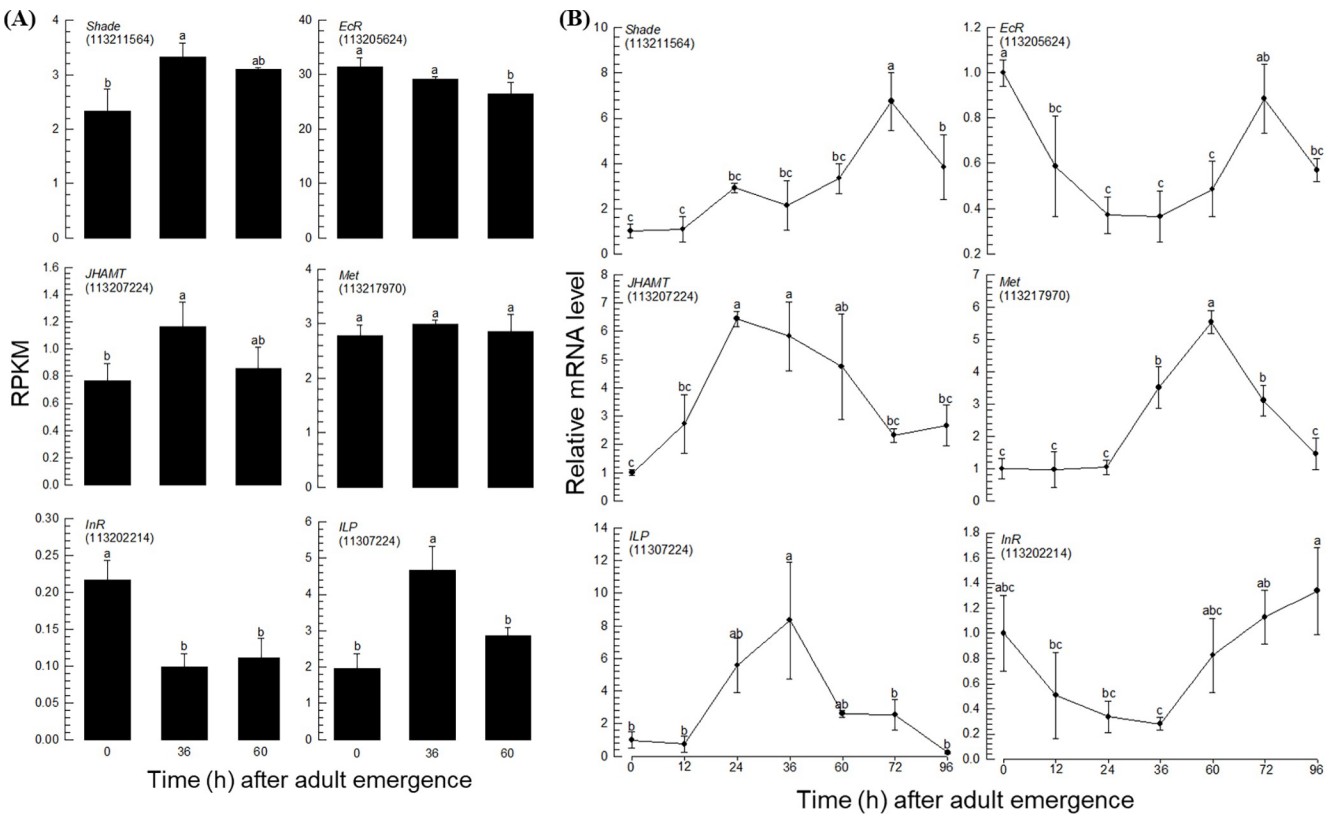

**Fig 5. Expression analysis of the selected endocrine signal genes during *F. occidentalis* ovarian development.** The NCBI Gene ID is in each parenthesis. (A) RNASeq analysis. (B) RT-qPCR analysis. Each measurement was replicated 3 times. The figures in parentheses indicate LOCs, which are the NCBI gene IDs. Different letters above the standard deviation bars represent significant differences among the means at type I error = 0.05 (DMRT test).

provided new insight into the evolution of insect ovaries. Later, tubuliferan thrips also showed germ cell clusters, indicating that the neo-panoistic ovariole-type prevailed in Thysanoptera [17]. This was supported by our current study using *F. occidentalis* ovarioles, which did not have nurse cells in the follicles, while a germ cell cluster was found in the germarium.

The ovaries of *F. occidentalis* grew just after adult emergence. During this period, the ovariole size increased along with oocyte development, and the final follicles in each ovariole had chorionated oocytes. Oogenesis is a sequential process consisting of previtellogenic development, vitellogenesis, and choriogenesis [18]. Previtellogenic development occurs in the germarium at the distal part of each ovariole and forms oocytes from the oogonial stem cells by mitosis and meiosis. Vitellogenesis is the process of accumulating vitellogenin (Vg) and other biomaterials into growing oocytes. After oocytes are fully grown, they are coated with chorion proteins secreted from the follicular epithelium to become eggs at the proximal part of the ovarioles. These eggs are then ovulated to the oviducts and fertilized just before oviposition. This ovarian developmental scenario suggests a sequence of oogenesis events in the neo-panoistic ovariole of *F. occidentalis*. First, primary oocytes may be produced from germ cell clusters in the germarium. Second, the oocytes grow in size by accumulating Vg during vitellogenesis. Finally, the follicular epithelium forms the chorion of the fully grown oocytes during choriogenesis. Our transcriptome analysis supported the oogenesis processes by providing expression profiles of *Vg*, *chorion proteins*, *mucin*, and *yellow* genes during oogenesis.

RNASeq analysis used the NovaSeq platform, which sequenced more than 100 million reads per sample and resulted in more than 80% gene mapping rates. The first draft genome of

*F. occidentalis* was reported and 16,859 genes were annotated [8]. Our RNASeq data from three ovarian developmental stages were mapped to 15,083 genes. Most of the mapped genes were shared among the three ovarian developmental stages. DEG analysis identified 473 and 443 DEGs in the mid and late stages, respectively. These DEGs were associated with metabolic pathways, especially related to nucleic acid biosynthesis and cofactor and amino acid biosynthetic pathways. The findings suggest that ovarian development in *F. occidentalis* requires a massive supply of raw materials such as nucleic acids and proteins.

Egg proteins were selected from the transcriptomes of the ovarian developmental stages. These genes represented highly expressed genes because they were increased more than 8-fold compared to the early ovarian developmental stage. The genes included *mucin* and *yellow* in addition to the well-known *Vg* and *chorion protein* genes. Mucins are high molecular and heavily glycosylated proteins with tandem repeats of identical or highly similar sequences rich in Ser, Thr, and Pro [19]. In insects, intestinal mucin is a major protein in the midgut peritrophic membrane, which facilitates the digestive process as well as protects the gut epithelium from microbial infections [20]. Salivary gland mucin might modulate the lubrication of insect mouthparts or defend plant attachment by inducing plant cell death through the formation of salivary sheaths [21, 22]. A specific mucin protein is known to be associated with the formation of eggshells in *Nilaparvapa lugens* and *Spodoptera exigua* [23, 24]. In our current study, the *mucin* gene highly expressed during *F. occidentalis* ovarian development suggests its function in chorion formation. Yellow and related major royal jelly protein (MRJP)-like proteins are widely found in insect genomes and these genes are classified into ten clades including Yellow-b, -c, -d/e3, -e, -f, -g/g2, -h, -y, -x and MRJP-like protein [25]. In the *Aedes albopictus* mosquito, Yellow-g and Yellow-g2 are localized in the exochorion and outer endochorion, where they mediate darkening processes to physically strengthen the chorions [26]. Thus, the high expression of the *yellow* gene during *F. occidentalis* ovarian development suggests its function in chorion formation.

Transcriptome analysis also showed the expression profiles of genes associated with endocrine signals during *F. occidentalis* ovarian development. In insects, different endocrine signals are associated with ovarian development. JH is a sesquiterpenoid that mediates a status quo effect during the immature stage to prevent precocious metamorphosis. However, in adults, it stimulates ovarian development as a gonadotropin in various insects [27]. JH directly stimulates Vg biosynthesis in some insects and facilitates Vg uptake by growing oocytes by inducing follicular patency [28]. In mosquito females, 20-hydroxyecdysone acts as a gonadotropin [29]. ILPs are known to mediate ovarian development by stimulating oogonial proliferation to produce oocytes in the stem cell niche located in the germarium of the distal ovariole [30]. In *F. occidentalis*, JH and ecdysteroid play crucial roles in mediating metamorphosis. Krüppel homolog 1 (*Kr-h1*) and Broad (*Br*) are transcription factors leading to larval and pupal characteristics under JH and ecdysteroid hormones, respectively [31, 32]. In *F. occidentalis*, Kr-h1 mRNA levels were high in the embryonic stage, remained at a moderate level in the larval and prepupal stages, and were low in the pupal stage. In contrast, *Br* mRNA levels were moderate in the embryonic stage and high at the larva-pupa transition stage. Except for *Br* expression in the embryonic stage, these two gene expression patterns followed the corresponding profiles of holometamorphic insects [33]. Furthermore, the adult specifier, *E93*, expression increased during immature development and its inhibition prevented adult metamorphosis [34]. However, little is known about JH and ecdysteroid mediation in oogenesis in *F. occidentalis*. Our current transcriptome analysis during ovarian development suggests that these endocrine signals play crucial roles in mediating *F. occidentalis* oogenesis based on their expression profiles. Increases in *ILP* and *JHAMT* expression at the mid ovarian developmental stage suggest their mediation of previtellogenesis by providing new oocytes from stem cells and vitellogenesis by

stimulating Vg synthesis and uptake. Maintaining high levels of *Shade* expression suggest a high level of ecdysteroid during ovarian development, which may stimulate metabolic pathways, especially protein and nucleic acid biosynthesis, in addition to stimulating Vg synthesis with the cooperation of JH.

This study reports the comparative transcriptomes of *F. occidentalis* during different stages of ovarian development. Although the transcriptome analyses do not completely represent the protein expression profiles, they gave us valuable insights on the thrips reproduction. The results suggest an increase in metabolic pathways along with protein and nucleic acid biosynthesis. The high upregulation of egg proteins such as Vg, chorion protein, and sclerotizing agents during choriogenesis was also found. Finally, JH, ecdysteroid, and insulin signals may play crucial roles in mediating *F. occidentalis* oogenesis. A recent study showed that prostaglandin mediates oocyte devilment in early and late stages in addition to the endocrine signals [35]. This suggests the oogenesis of *F. occidentalis* would be a model system for an integrative analysis of endocrine signals mediating different reproductive processes of previtellogenesis, vitellogenesis, and choriogenesis.

## Supporting information

**S1 Table. Primers used in this study.**
(DOCX)

**S2 Table. Annotation of 53 genes expressed only at 36 h after adult emergence (AAE) compared to expression levels at the early (0 h AAE) developmental stage in female *F. occidentalis* adults.**
(DOCX)

**S3 Table. Annotation of 68 genes expressed only at 60 h after adult emergence (AAE) compared to expression levels at the early (0 h AAE) developmental stage in female *F. occidentalis* adults.**
(DOCX)

**S4 Table. Highly (> 8-fold) suppressed genes at mid (36 h after adult emergence) and late (60 h after adult emergence) ovarian development stages compared to expression levels in the early (0 h after adult emergence) developmental stage in female *F. occidentalis* adults.**
(DOCX)

## Acknowledgments

We thank Chulyoung Kim in our laboratory to provide thrips for our analysis.

## Author Contributions

**Conceptualization:** Yonggyun Kim.

**Data curation:** Du-Yeol Choi.

**Formal analysis:** Du-Yeol Choi.

**Funding acquisition:** Yonggyun Kim.

**Investigation:** Du-Yeol Choi, Yonggyun Kim.

**Methodology:** Du-Yeol Choi.

**Project administration:** Yonggyun Kim.

**Resources:** Yonggyun Kim.

**Software:** Du-Yeol Choi.

**Supervision:** Yonggyun Kim.

**Validation:** Du-Yeol Choi.

**Visualization:** Du-Yeol Choi.

**Writing – original draft:** Du-Yeol Choi, Yonggyun Kim.

**Writing – review & editing:** Yonggyun Kim.

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
