## [Decision Letter · Decision Letter 0]

30 Jun 2022

PONE-D-22-15697Transcriptome analysis of female western flower thrips, Frankliniella occidentalis, exhibiting neo-panoistic ovarian developmentPLOS ONE

Dear Dr. Kim,

Thank you for submitting your manuscript to PLOS ONE. After careful consideration, we feel that it has merit but does not fully meet PLOS ONE’s publication criteria as it currently stands. Therefore, we invite you to submit a revised version of the manuscript that addresses the points raised during the review process. Please submit your revised manuscript by Aug 14 2022 11:59PM. If you will need more time than this to complete your revisions, please reply to this message or contact the journal office at plosone@plos.org. Please include the following items when submitting your revised manuscript:A rebuttal letter that responds to each point raised by the academic editor and reviewer(s). You should upload this letter as a separate file labeled 'Response to Reviewers'.A marked-up copy of your manuscript that highlights changes made to the original version. You should upload this as a separate file labeled 'Revised Manuscript with Track Changes'.An unmarked version of your revised paper without tracked changes. You should upload this as a separate file labeled 'Manuscript'.If applicable, we recommend that you deposit your laboratory protocols in protocols.io to enhance the reproducibility of your results. Protocols.io assigns your protocol its own identifier (DOI) so that it can be cited independently in the future. For instructions see: https://journals.plos.org/plosone/s/submission-guidelines#loc-laboratory-protocols. Additionally, PLOS ONE offers an option for publishing peer-reviewed Lab Protocol articles, which describe protocols hosted on protocols.io. Read more information on sharing protocols at https://plos.org/protocols?utm_medium=editorial-email&utm_source=authorletters&utm_campaign=protocols.

We look forward to receiving your revised manuscript.

Kind regards,

Academic Editor

PLOS ONE

Journal Requirements:

Reviewers' comments:

Reviewer's Responses to Questions

**Comments to the Author**

1. Is the manuscript technically sound, and do the data support the conclusions?

Reviewer #1: Yes

Reviewer #2: Yes

2. Has the statistical analysis been performed appropriately and rigorously? 

Reviewer #1: Yes

Reviewer #2: N/A

3. Have the authors made all data underlying the findings in their manuscript fully available?

Reviewer #1: Yes

Reviewer #2: Yes

4. Is the manuscript presented in an intelligible fashion and written in standard English?

Reviewer #1: Yes

Reviewer #2: Yes

5. Review Comments to the Author

Reviewer #1: Reviewer’s comment

Title: Transcriptome analysis of female western flower thrips, Frankliniella occidentalis, exhibiting neo-panoistic ovarian development.

In this study, the authors Choi and Kim have done transcriptome analysis of pest western flower thrips Frankliniella occidentalis at mid (36 h after adult emergence (AAE)) and late (60 h AAE) ovarian developmental stages using RNA sequencing (RNASeq) technology. More than 120 million reads per replication were matched to » 15,000 F. occidentalis genes. Almost 500 genes were expressed at each mid and late ovarian developmental stage. Differentially expressed genes (DEGs) were associated with metabolic pathways and protein and nucleic acid biosynthesis. In both ovarian developmental stages, vitellogenin, mucin, and chorion genes were highly (> 8-fold) expressed. Endocrine signals associated with ovarian development were further investigated from the DEGs. Insulin and juvenile hormone signals were upregulated only at 36 h AAE, whereas the ecdysteroid signal was highly maintained at 60 h AAE.

The thrip is the primary vector for a few plant viruses such as tomato spotted wilt virus (TSWV) and impatiens necrotic spot virus (INSV). The study is interesting and vital as the thrips can destroy many crops worldwide by feeding on them and by spreading the plant viruses. There are few reports on the transcriptomic analysis of the thrips during virus infection, but that of on reproduction of F. occidentalis is rare in the literature. However, the discussion part is important here. The RNA/gene expression results are straight forward but how these differentially expressed genes contribute to a faster reproduction rate needs to be discussed in detail. Here are few suggestions which might improve the value of the manuscript.

1. Discussion: How these results help in the control of the pest directly or indirectly needs to be explained. What is the significance of expressed genes? The authors may consider including literature support on this aspect.

2. Resolution of figures 3, 4 and 5 to be improved.

3. Line 98-99: Mention that the total RNA extracted was from the whole thrips.

4. As this article is on the reproduction of thrips and hence, more detail related to reproduction can be included in the introduction. How fast the pest can develop into an adult (Moritz et al. 2004. Virus Res., 100 pp. 143-149), etc.

5. The transcriptomics results need not necessarily correlate to protein expression (PMID: 26085669). Authors may consider including it in the discussion.

Reviewer #2: The article by Choi et al. summarizes ovarian development and associated transcriptomes of F. occidentalis at various stages of oocyte development (0 h AAE, 36 h AAE, and late 60 h AAE) and compares the differentially expressed genes. The study is technically well performed and described in the manuscript, yet there are some concerns that should be addressed before it can be reconsidered for publication.

1. Why do authors choose to perform RNA extraction from whole female adult F. occidentalis instead of just from the ovarioles?

2. In Figure 1, the authors have only shown one stage (3-day-old female adults) of ovariole structure. Since the authors are focusing on all three stages ( 0h, 36 h, and 60 h AAE) and are also measuring the ovariole length, it is important to show all stages of ovariole development.

3. For Figure 2B, it might be better to include volcano plots with differentially expressed genes pattern (36 vs 0, 60 vs 0, 36 vs 60) and top candidate gene names to guide readers to follow the changes in gene expression.

4. In Figure 3, it is difficult to follow the entire stretch of the graph. Importantly, the labels are not clearly legible. Authors can show only the pathways that are up or down-regulated between 60 vs 36 instead of showing the number of genes.

5. Figures 4 & 5: Authors may consider merging these two figures into one figure. Also, is there a particular reason for including RPKM values on the main figure?

6. PLOS authors have the option to publish the peer review history of their article (what does this mean?). If published, this will include your full peer review and any attached files.

Reviewer #1: No

Reviewer #2: **Yes: **Prasanna Katti

---

## [Author Response · Author response to Decision Letter 0]

3 Jul 2022

[Reviewer #1]

In this study, the authors Choi and Kim have done transcriptome analysis of pest western flower thrips Frankliniella occidentalis at mid (36 h after adult emergence (AAE)) and late (60 h AAE) ovarian developmental stages using RNA sequencing (RNASeq) technology. More than 120 million reads per replication were matched to » 15,000 F. occidentalis genes. Almost 500 genes were expressed at each mid and late ovarian developmental stage. Differentially expressed genes (DEGs) were associated with metabolic pathways and protein and nucleic acid biosynthesis. In both ovarian developmental stages, vitellogenin, mucin, and chorion genes were highly (> 8-fold) expressed. Endocrine signals associated with ovarian development were further investigated from the DEGs. Insulin and juvenile hormone signals were upregulated only at 36 h AAE, whereas the ecdysteroid signal was highly maintained at 60 h AAE.

The thrip is the primary vector for a few plant viruses such as tomato spotted wilt virus (TSWV) and impatiens necrotic spot virus (INSV). The study is interesting and vital as the thrips can destroy many crops worldwide by feeding on them and by spreading the plant viruses. There are few reports on the transcriptomic analysis of the thrips during virus infection, but that of on reproduction of F. occidentalis is rare in the literature. However, the discussion part is important here. The RNA/gene expression results are straight forward but how these differentially expressed genes contribute to a faster reproduction rate needs to be discussed in detail. Here are few suggestions which might improve the value of the manuscript.

Comment #1-1: Discussion: How these results help in the control of the pest directly or indirectly needs to be explained. What is the significance of expressed genes? The authors may consider including literature support on this aspect.

Response: A main issue of this study was to understand molecular processes underlying oogenesis of the thrips. Thus the appropriate discussion related with this issue is added by providing a usefulness of the results with an additional reference as follows: “A recent study showed that prostaglandin mediates oocyte devilment in early and late stages in addition to the endocrine signals [35]. This suggests the oogenesis of F. occidentalis would be a model system for an integrative analysis of endocrine signals mediating different reproductive processes of previtellogenesis, vitellogenesis, and choriogenesis.”

Comment #1-2: Resolution of figures 3, 4 and 5 to be improved.

Response: These figures are redrawn to improve resolution and replaced.

Comment #1-3: Line 98-99: Mention that the total RNA extracted was from the whole thrips.

Response: Added and rephrased as follows: “Total RNAs were extracted from the whole bodies of female F. occidentalis at different ages (0, 36, and 60 h after adult emergence).”

Comment #1-4: As this article is on the reproduction of thrips and hence, more detail related to reproduction can be included in the introduction. How fast the pest can develop into an adult (Moritz et al. 2004. Virus Res., 100 pp. 143-149), etc.

Response: We add the following characters related with the serious issue on reproduction and outbreak: “A brief immature period less than 10 days along with this various reproductive modes allow the thrips to rapidly build up the field populations during crop cultivating periods and so frequently leads to outbreaks beyond economic injury level [6].”

Comment #1-5: The transcriptomics results need not necessarily correlate to protein expression (PMID: 26085669). Authors may consider including it in the discussion.

Response: We agree on this comment. We add the following statement in the discussion: “Although the transcriptome analyses do not completely represent the protein expression profiles, they gave us valuable insights on the thrips reproduction.”

 

[Reviewer #2]

The article by Choi et al. summarizes ovarian development and associated transcriptomes of F. occidentalis at various stages of oocyte development (0 h AAE, 36 h AAE, and late 60 h AAE) and compares the differentially expressed genes. The study is technically well performed and described in the manuscript, yet there are some concerns that should be addressed before it can be reconsidered for publication.

Comment #2-1: Why do authors choose to perform RNA extraction from whole female adult F. occidentalis instead of just from the ovarioles?

Response: We understand the issue raised by the reviewer. The best option was to use the isolate ovary samples. However, whole body samples give additional information probably from transcriptomes of fat body and hemolymph, which are the tissues associated with the reproduction. To be clear, we added “the whole body isolation” in the materials and methods. 

Comment #2-2: In Figure 1, the authors have only shown one stage (3-day-old female adults) of ovariole structure. Since the authors are focusing on all three stages ( 0h, 36 h, and 60 h AAE) and are also measuring the ovariole length, it is important to show all stages of ovariole development.

Response: We agree on the issue raised by the reviewer. Thus, we showed the ovarian development from 0 day to 7 days after adult emergence in Fig. 1B. The photo demonstrates the entire ovary structure of this species.

Comment #2-3: For Figure 2B, it might be better to include volcano plots with differentially expressed genes pattern (36 vs 0, 60 vs 0, 36 vs 60) and top candidate gene names to guide readers to follow the changes in gene expression.

Response: Main points are the specific genes at different reproductive stages. Supplementary Tables separately indicate the genes specific to each development: 

Table S2. Annotation of 53 genes expressed only at 36 h after adult emergence (AAE) compared to expression levels at the early (0 h AAE) developmental stage in female F. occidentalis adults

Table S3. Annotation of 68 genes expressed only at 60 h after adult emergence (AAE) compared to expression levels at the early (0 h AAE) developmental stage in female F. occidentalis adults

Table S4. Highly ( > 8-fold) suppressed genes at mid (36 h after adult emergence) and late (60 h after adult emergence) ovarian development stages compared to expression levels in the early (0 h after adult emergence) developmental stage in female F. occidentalis adults

Comment #2-4: In Figure 3, it is difficult to follow the entire stretch of the graph. Importantly, the labels are not clearly legible. Authors can show only the pathways that are up or down-regulated between 60 vs 36 instead of showing the number of genes.

Response: We separate the figure by moving the each KEGG category to figure caption. 

Comment #2-5: Figures 4 & 5: Authors may consider merging these two figures into one figure. Also, is there a particular reason for including RPKM values on the main figure?

Response: These two figures explain the different aspects of the oogenesis. Fig. 4 indicates specific egg proteins and their expression profiles. Fig. 5 indicates the underlying endocrine signals. RPKM values along with detailed RT-qPCR support the validation of the RNASeq.

---

## [Decision Letter · Decision Letter 1]

20 Jul 2022

Transcriptome analysis of female western flower thrips, Frankliniella occidentalis, exhibiting neo-panoistic ovarian development

PONE-D-22-15697R1

Dear Dr. Kim,

We’re pleased to inform you that your manuscript has been judged scientifically suitable for publication and will be formally accepted for publication once it meets all outstanding technical requirements.

Kind regards,

Academic Editor

PLOS ONE

Additional Editor Comments (optional):

Reviewers' comments:

Reviewer's Responses to Questions

**Comments to the Author**

1. If the authors have adequately addressed your comments raised in a previous round of review and you feel that this manuscript is now acceptable for publication, you may indicate that here to bypass the “Comments to the Author” section, enter your conflict of interest statement in the “Confidential to Editor” section, and submit your "Accept" recommendation.

Reviewer #1: (No Response)

Reviewer #2: All comments have been addressed

2. Is the manuscript technically sound, and do the data support the conclusions?

Reviewer #1: Partly

Reviewer #2: Yes

3. Has the statistical analysis been performed appropriately and rigorously? 

Reviewer #1: I Don't Know

Reviewer #2: N/A

4. Have the authors made all data underlying the findings in their manuscript fully available?

Reviewer #1: No

Reviewer #2: Yes

5. Is the manuscript presented in an intelligible fashion and written in standard English?

Reviewer #1: Yes

Reviewer #2: Yes

6. Review Comments to the Author

Reviewer #1: The author's responses are satisfactory. However I could not locate the revised figures. Not sure if there is any technical error?

Reviewer #2: (No Response)

7. PLOS authors have the option to publish the peer review history of their article (what does this mean?). If published, this will include your full peer review and any attached files.

Reviewer #1: No

Reviewer #2: No

---

## [Editor Report · Acceptance letter]

22 Jul 2022

PONE-D-22-15697R1 

Transcriptome analysis of female western flower thrips, *Frankliniella occidentalis*, exhibiting neo-panoistic ovarian development  

Dear Dr. Kim:

I'm pleased to inform you that your manuscript has been deemed suitable for publication in PLOS ONE. Congratulations! Your manuscript is now with our production department. 

Kind regards, 

on behalf of

Dr. Rajakumar Anbazhagan 

Academic Editor

PLOS ONE